# Optimal Deployment of WSN Nodes for Crop Monitoring Based on Geostatistical Interpolations

**DOI:** 10.3390/plants11131636

**Published:** 2022-06-21

**Authors:** Edgar Andres Gutierrez, Ivan Fernando Mondragon, Julian D. Colorado, Diego Mendez Ch

**Affiliations:** 1School of Engineering, Pontificia Universidad Javeriana, Bogota 110231, Colombia; edgar-gutierrez@javeriana.edu.co (E.A.G.); diego-mendez@javeriana.edu.co (D.M.C.); 2School of Electronics Engineering, Universidad Santo Tomás Colombia, Bogota 150001, Colombia

**Keywords:** WSN, remote sensing, soil moisture, Kriging interpolation, potato crops

## Abstract

This paper proposes an integrated method for the estimation of soil moisture in potato crops that uses a low-cost wireless sensor network (WSN). Soil moisture estimation maps were created by applying the Kriging technique over a WSN composed of 11×11 nodes. Our goal is to estimate the soil moisture of the crop with a small-scale WSN. Using a perfect mesh approach on a potato crop, experimental results demonstrated that 25 WSN nodes were optimal and sufficient for soil moisture characterization, achieving estimations errors <2%. We provide a strategy to select the number of nodes to use in a WSN, to characterize the moisture behavior for spatio-temporal analysis of soil moisture in the crop. Finally, the implementation cost of this strategy is shown, considering the number of nodes and the corresponding margin of error.

## 1. Introduction

High-throughput crop phenotyping is strictly related to the spatio-temporal characterization of several plant variables in response to both biotic and abiotic factors. According to [1], it is clear that a high soil moisture content must be maintained at all stages of plant growth to obtain high potato yields. Soil moisture should never be allowed to fall below 50% of the available moisture range. Due to its shallow root system, the potato is sensitive to drought [2]. Predicting soil moisture in potato crops requires as input the initial crop condition prior to any irrigation regime. The techniques and impacts of irrigation on growth and yield have been extensively analyzed by Djaman et al. [3]. To this end, wireless sensor networks (WSNs) have recently gained significant traction, allowing redundant and precise measurements for improving crop management practices. Several WSN-driven applications are focused on characterizing soil moisture and temperature [4,5,6], relative air humidity [7], electrical conductivity [8], pH [9] and wind speed and direction [10]. These variables have been widely analyzed in research related to crops such as lavender, mint [11], walnut [12], cherry [13], rice [14], tomatoes, lettuce [9], corn [8,15,16], lemon [17], potatoes [10] and grapes [18].

In order to improve on the spatio-temporal analysis of the crop variables, several works have proposed the use of WSNs with numerical interpolation methods found in geostatistical applications [19,20,21]. In this regard, the well-known Kriging technique is being applied for achieving continuous data gradients for accurate crop mapping. Considering the need to manage water resources to improve on crop yield, the work in [22] proposed an IoT architecture composed by 104 WSN nodes for the real-time sensing of soil moisture and water stress. Optimization methods were applied aimed at reducing the number of nodes while also reducing the energy consumption of the entire WSN. Other works reported in [23,24,25,26,27] also present optimal approaches for the spatial distribution of WSN nodes, considering several properties of the soil and irrigation systems, among other variables.

In [28], a hydro-agricultural system was developed based on the operation of a WSN driven by the Kriging technique. The proposed system allows the interpolation of the near-surface soil moisture of the crop with aerial spectral imaging data of the canopy. Following the aforementioned approach, the works in [29,30] also presented a soil irrigation management system combining co-Kriging and Kriging with external drift (KED) techniques, aimed at optimizing the spatial distribution of the WSN. Besides the interpolation methods, the authors in [31] mentioned the importance of acquiring accurate and relevant data that enable the extraction of relevant features for training and scaling up estimation methods via data interpolation or machine learning models. In this regard, Monte Carlo simulations have been used to randomly distribute WSN nodes, and for minimizing the variance of the sensor readings, while allowing an optimal distribution for acquiring the ground-truth data [32,33,34].

Previous research using the Kriging interpolation method for soil moisture estimation used many hectares of land, and even states of a country. However, these solutions are economically unfeasible for small farmers, who require continuous information on their crops, and the technology implemented for these cases is too expensive. As for the studies carried out on a small scale, it has been observed that the nodes are randomly placed within the crop, almost resembling the methodologies of soil sampling for a crop, which is not correct, since the objective of a soil study is different from checking the state of soil moisture and analyzing it from a spatio-temporal point of view.

In this paper, we propose an optimal approach for WSN deployment that uses the Kriging interpolation technique in order to estimate the spatial moisture for potato crops *Solanum phureja*. In-field experimental trials were conducted to demonstrate how to minimize the number of WSN nodes without compromising the accuracy of soil moisture estimation. Our contribution is twofold: (i) an integrated low-cost and small-scale WSN that can scale up depending on crop mapping requirements, (ii) a methodology to optimally deploy the WSN, by drastically reducing the number of sensing nodes; for the application at hand, an initial WSN composed of 289 nodes was reduced to 25 nodes, while maintaining estimation error of <2%. The forthcoming sections will introduce the proposed Kriging-driven WSN architecture, the methodology for optimal deployment, and the experimental results aimed at crop irrigation management.

## 2. Materials and Methods

The architecture proposed in Figure 1 shows the five stages of this project: (A) selection of the study area and soil moisture measurement, (B) filtering data, (C) referencing the Kriging map, (D) distribution of the perfect grid and construction of the map for n×n nodes and (E) calculation of error. Below is a description of these stages:

The potato test crop is located in the municipality of Monguí, in the region of Boyacá, Colombia (5°42′56.5″ N, 72°50′39″ W). The region has an average altitude of 2900 m.a.s.l., an average annual temperature of 13 °C and average annual rainfall of 935 mm. The crop has an area of 64 m^2^ (8 m × 8 m). Figure 2 shows the initial Cartesian XY plane made for the crop. A grid was built with 0.5 m × 0.5 m squares. Capacitive analog sensors to measure soil moisture were installed in each of the 289 grid intersection points. Soil moisture measurements were made between days 93 and 107 of the year (rainy season in Colombia). The average time interval in which soil moisture measurements were taken was 15 min. The information collected from the 289 measurement points allowed constructing the initial soil moisture map.

### 2.1. Study Area

Each grid space was planted with one potato seed of the *Solanum phureja* variety (256 plants in total). During the planting period, a mixture of mineral fertilizers known as NPK was applied to meet the nutritional needs of the plants. The composition of the fertilizer applied is: phosphorus (15%), nitrogen (15%) and potassium (15%). Nitrogen stimulates the growth of leaves and branches, and phosphorus stimulates root growth and flowering. Potassium is essential for fruit, seed and tuber growth. The composition of the soil in the test crop was: clay [18], loamy [35] and sandy soils [17]. The characteristics and properties of the soil can affect the data provided by the sensors. It is important to keep in mind that in traditional crops, the soil is not homogeneous, since weeds, rocks, organic and inorganic residues can be found, which can alter the capacity of the soil to absorb moisture and the distribution of the crop. An analysis of the soil was conducted in the laboratory, and it determined that it is within the *loamy* category.

#### Measurement Equipment

To obtain humidity data, WSN nodes were constructed mainly with Arduino Nano development boards. See Figure 3. The type of the soil moisture sensor was analog capacitive, manufactured by the company DFRobot. A reference voltage of 5 V is required to perform the ADC inside the microcontroller. The microcontroller converts the analog signals into humidity percentages (0–100%).

The gravimetric method was used to calibrate the sensors. An initial sample of the soil with weight ($S_{d}$) of 680 g from the crop was selected and then dried in an oven. The moisture sensor was kept inside the sample at all times, and a record of the values delivered by a 10 bit ADC was kept. From this point on, the weight of the wet soil ($S_{w}$) was recorded after applying 5 mL of water every 30 s. This process was carried out until the sample was saturated and reached 832 g. The soil moisture percentage ($S_{p}$) was calculated by applying the following equation:(1)$$S_{p}=\left(\frac{S_{w}-S_{d}}{S_{d}}\right)*100,$$

By obtaining $S_{p}$ and the ADC value, the calibration curve for the sensor was obtained. Figure 4 shows the calibration function $f\left(x\right)=\left(4.61\times 10^{6}x^{-3.044}\right)*100$. Soil moisture is shown as 0% to 100%: 0% indicates extreme dryness and 100% total wetness or saturation of the crop soil.

To facilitate mobility and data collection, wireless communication via a 802.15.4 compliant radio (including the ZigBee stack) was chosen for each node. This information was centralized in a PC and organized in excel for further processing. As for the power supply of each of the nodes, a 5 V/1 A output solar energy manager was used to power the microcontroller, in addition to a solar panel with a 5V/1A regulator and a 3.7 V/900 mAh lipo battery. The measurements in the *Solanum phureja* potato crop were made between days 93 and 107 of the year (rainy season in Colombia). A total of 289 points referenced in an XY plane were taken in the 64 m^2^ crop. See Figure 5.

The information collected from the 289 sensors related to soil moisture measurements from the crop was stored in a matrix and then processed in the next stage.

### 2.2. Filtering

When a low-cost humidity sensor is used to acquire samples, errors may occur. For this reason, data smoothing was applied using frequency domain filters. At this stage, a two-dimensional Gaussian low-pass filter was implemented, as defined in the following equation:(2)$$H\left(u,v\right)=e^{-D^{2}\left(u,v\right)/2\sigma^{2}},$$

*D*(*u*,*v*) is the distance in relation to the frequency. By making $\sigma =D_{0}$, we can express the filter in the notation:(3)$$H\left(u,v\right)=e^{-D^{2}\left(u,v\right)/2D_{0}^{2}},$$

A value of $D_{0}=15$ was used because it smooths the data without affecting the amplitude value and the mean value of the map. If you modify the $D_{0}$ by very low values ($D_{0}<8$), it will affect the amplitude of the data and will generate serious errors in the subsequent analysis. Note that $D\left(u,v\right)=D_{0}$. It is important to remember that the inverse Fourier transform of a Gaussian filter is a Gaussian function.

### 2.3. Reference Kriging Maps

To conduct the analysis of the spatial variability of soil moisture, the mean ($\bar{Z}$), the standard deviation (σ), the variance ($s^{2}$), the coefficients of variation (C.V), the maximum ($Z_{max}$) and minimum ($Z_{min}$) values were calculated, once the 289 moisture measurements ($Z_{i}$) on the potato crop grid were completed.

Soil moisture within a potato crop presents temporal and spatial variations. Since it was not possible to use sensors throughout the entire crop, soil moisture estimation techniques were used by interpolating the data measured at specific points referenced in a Cartesian plane. There are several methods from geostatistics that are useful for obtaining such interpolation, taking into account the locations of the sensors in an *XY* plane.

For the present research, the Kriging method was used as an interpolation technique to analyze the moisture of the soil. Before Kriging is used, and for it to be efficient, the soil moisture data must follow a normal distribution (cdf). Equation (Equation 4) shows a classic method to verify that the type of distribution of the moisture data runs through the P–P normal probability plot. After applying the equation, a straight line should be generated if the data complies with the normal distribution.
(4)$$cdf=\frac{1}{2}\left[1+erf\left(\frac{Z_{i}-\bar{Z}}{\sigma \sqrt{2}}\right)\right],$$

Subsequently, Kriging relies on variography structural analysis by characterizing the spatial continuity of a dataset within the crop. The variogram model includes techniques such as the Gaussian model, the spherical model and the double spherical model. Several research initiatives determined that one of the best models is the variogram based on the double spherical model, and it was the one utilized for the present study. To consider all possible locations in *x*, the semivariogram $\gamma \left(h\right)$ was used for the lag distance *h*, which is defined by Equation (Equation 5).
(5)$$\gamma \left(h\right)=\frac{1}{2N\left(h\right)}\sum_{i=1}^{N\left(h\right)}{\left[Z\left(x_{i}+h\right)-Z\left(x_{i}\right)\right]}^{2},$$
where *h* is the delay distance, *N*(*h*) is the number of pairs for the *h* delay and *Z*(*x*) is the soil moisture value at location *x*. The typical variogram response in *x*-direction is shown in Figure 6.

The range represents the distance for which there is no correlation between the measured soil moisture data points; the nugget represents a minimum variation; and the contribution, also known as sill, determines the average variance of points at a distance where there is no longer a correlation between points. In conclusion, the analysis using variograms allows us to know the maximum distance of the related data.

To estimate the surface area utilizing the deployed nodes, the semivariogram was used as part of the Kriging interpolation. After analyzing the results of the semivariogram, optimal weights were assigned to the known values at the measured points (*x*,*y*) to estimate the values at the unknown points. These weights depend on the value of the known sample and the distribution shape of the soil moisture data in the crop. The goal of the Kriging technique, through its interpolation, is to choose the optimal weights by comparing the minimum estimation error. The estimated value of an unknown point ($E(x_{i},y_{j})$) can be estimated by calculating the weighted sum of the known points, as presented in Equation (Equation 6).
(6)$$E\left(x_{i},y_{j}\right)=\sum_{k=1}^{k=K}\omega_{k}z_{k},$$
where *K* is the set of points with known coordinates $\left(x_{i},y_{i}\right)$ and with soil moisture value $z_{i}$; $\omega_{k}$ is the weight given to the known points. Weights were calculated using a set of weights $\omega_{ij}$ for each estimation. To calculate the set of weights $\omega_{ij}$, a variogram *V* was constructed from the known points, represented by Equation (Equation 7), in which $D_{ij}$ is taken as the distance between known pair of points (from one known point $\left(x_{i},y_{i}\right)$ to another known point $\left(x_{j},y_{j}\right)$).
(7)$$V=\left[\begin{array}{ccc} D_{11} & \cdots & D_{1k}\\ \vdots & \ddots & \vdots \\ D_{k1} & \cdots & D_{kk} \end{array}\right],$$

Finally, the vector of weights for the unknown values can be calculated by Equation (Equation 8):(8)$$\omega_{ij}=d_{ij}V^{-1},$$

Considering the vector of weights of the unknown values, Equation (Equation 6) is used again to calculate the estimated soil moisture values for these points and generate the soil moisture distribution map of the potato crop.

#### Model Used in Kriging

Regarding Kriging, several models can be used in terms of variograms that will allow better describing the behavior of the variable compared to the distance. Various working models are used to generate the variogram; among these are the spherical, the pentaspherical, the bounded linear, the circular, the exponential and the Gaussian. For this research, the Gaussian model was selected, which uses a normal probability distribution curve. This type of model is useful when the phenomena are similar at short distances due to their progressive ascent on the *y*-axis.
(9)$$\gamma \left(h\right)=\left\{\begin{array}{cc} 0, & h=0\\ C_{0}+C\left(1-e^{\frac{-h^{2}}{a^{2}}}\right), & h>0 \end{array}\right.,$$
where *C* is the variance, $C_{0}$ is the nugget constant, *a* is the effective range and $C+C_{0}$ is the sill.

### 2.4. Perfect Grid Distribution and Map Construction for n×n Nodes and Error Calculation

After the filtering stage in which the signal noise was reduced and using a perfect grid of n×n nodes, new Kriging maps were generated, to be compared with the reference map. The reference crop soil moisture map, as described above, had a size of 8 m × 8 m and had been created from 289 locations. An equidistant distribution between the nodes of the n×n network of n=2,3,…,11 within the test crop was proposed for the new maps. This allowed us to calculate the error between the maps, using a point-to-point difference between the generated matrices, which can be seen in Figure 7.

Subsequently, Equation (Equation 10) is a more formal formulation of the calculated error between the reference map and the Kriging reconstruction:(10)$$e_{n*n}=\frac{\sum_{i=1}^{801}\left(\sum_{j=1}^{801}M_{Kr}\left(i,j\right)-M_{Kn*n}\left(i,j\right)\right)}{641601-\left(n*n\right)},$$
where $e_{n*n}$ is the error calculated from the difference between the reference soil moisture Kriging map $M_{Kr}$ and the Kriging map calculated for each of the $M_{Kn*n}$ grids of n×n number of nodes. It should be noted that the reference Kriging map uses 289 nodes, that is, a quantity of 17×17 nodes, due to the fact that it is an interpolation; it generates a matrix with a resolution of 801×801 for a total of 641.601 datapoints. For this reason, to make an adequate comparison, the dimensions of the resolution in quantity of data of the Kriging map for n×n nodes and the matrices should be the same.

## 3. Results and Discussion

### 3.1. Statistical Values of Soil Moisture

Table 1 presents the mean statistical values generated from the 12 experiments that were performed in the field.

The normal probability distribution of the data was verified while taking into account the theoretical and empirical probability values. It was represented in a linearization of the data expressed by f=0.9983X−0.005094 with $R^{2}=0.9972,SEE=0.06646$ and an adjusted R=0.9972. This linearization gives viability to the use of Kriging as an interpolation technique for soil moisture, as presented in Figure 8.

The histogram of the measured data follows a Gaussian distribution, reflecting the effect of the rainy season by maintaining large ranges of soil moisture in the crop, as Figure 9 shows.

### 3.2. Testing of the Proposed Architecture for Spatial Estimation of Soil Moisture

To validate the proposed architecture, 3 tests were conducted: (1) data without irrigation, (2) irrigation focused on the boundary of the crop and (3) irrigation focused on the center of the crop. Each one of these tests and their obtained results are shown below.

#### 3.2.1. Soil Moisture Analysis of the Crop without Irrigation

The initial test of the system obtained data from 289 measurement points without applying any type of irrigation. These measurement points were located within the crop, equidistant from each other (0.5 m), as shown in Figure 10.

Subsequently, the perfect grids were generated by taking into account the reference map. Table 2 shows the distances for each of the n×n configurations.

Considering these configurations and the proposed architecture, Kriging maps were calculated for each of these perfect grids, as Figure 11 presents.

Figure 11 shows the maps calculated from n=2 to n=11. As should be expected, the higher the number of nodes (n×n), the more similar they are to the reference map. It is important to evaluate the maximum and minimum points of each map. Visually, it is evident that n=2,3 and 4 do not present an important similarity to the reference map. Considering Equation (Equation 10), the error for each map of n×n nodes was calculated, and Figure 12 illustrates these errors.

Keeping in mind that soil moisture is in the range of 0% to 100%, the average error found for four nodes (2×2) was 5.12%. Subsequently, it became evident that as the number of nodes increased, the error decreased. For example, for 64 nodes (8×8) the error was 2.99%; for 121 nodes (11×11) it rose a little to 3.4%, a difference of 0.5%. This is not significant, since in subsequent tests it was found that the error kept decreasing for a larger number of nodes, with a steady trend to zero. Since the main goal of our study was to use as few sensors as possible, optimizing the general quality of the estimations to continue increasing the number of nodes was not considered convenient. Instead, error calculation was used to decide on the number of sensors to employ.

#### 3.2.2. Irrigation Focused on One Crop Boundary

As a second experiment, irrigation was carried out in one of the corners of the crop with a radial sprinkler, as presented in Figure 13. Again, the 289 measurements were taken, the data were smoothed to avoid noise in them, and as in the previous case, the soil moisture reference map was constructed. The irrigation area within the crop was not very large to analyze how the system responds to focused irrigation.

Similarly, maps were calculated for the n×n nodes, as Figure 14 shows.

As the area irrigated within the crop was not too large and because the error calculation equation was used, as it can be seen in Figure 15, the increase in moisture in a corner was not significant compared to the average value of the crop moisture. The soil moisture error for the n×n nodes ranged from a maximum value of 2.53% to a minimum of 1.1%. This was also because the crop was small and it did not represent a large area, which led to our next experimental test.

#### 3.2.3. Irrigation Focused on the Center of the Crop

A final test was conducted with a radial sprinkler irrigation aimed at the center of the crop, increasing the irrigation coverage area. The data from the 289 measurement points were also taken and passed through the data smoothing filter, and the reference map of soil moisture distribution was finally constructed (see Figure 16). It is necessary to remember that the tests were conducted on different days, since it was necessary to allow the amount of water applied in each of the tests to stabilize and absorb, so it would not affect the analyses sought after each irrigation.

Similarly, Kriging maps were calculated for the n×n nodes, as presented in Figure 17. It is interesting to see how the Kriging interpolations behaved; for instance, as the number of nodes in the test crop increased, the Kriging interpolations got closer and closer to the initial reference map. The same results were obtained in the previous tests, but it was easier to visually recognize this action for this particular case.

This is where we had to consider the extent to which the error could be tolerated. By calculating the errors of the n×n nodes for this test, we could see how the error effectively decreased as the number of nodes in the crop increased, as shown in Figure 18. When working on a flat terrain, it was evident that the Kriging interpolation decreased the error rate when calculating an average value for the crop, due to the interpolations.

The error of the maps for a WSN of 2×2 nodes was 4.5%, and it decreased for the case of 8×8 nodes, for which the error in relation to the reference map was 1.10%. After this point, the following errors remained at a margin very close to 1, because the distance variations between sensors was very short as the number of nodes increased. Keep in mind that for n=7,8,9,10 and 11, the distances are 1, 0.88, 0.8, 0.72 and 0.66 m, respectively. The distances for n=2,3,4,5,6 were a little larger, 2.6, 2, 1.6, 1.3 and 1.14 m. This became another important factor in determining the number of sensors to be used: for a small 8 m × 8 m flat crop like ours, it may not be as efficient to use a very large number of nodes, since the topography and the area will not allow much traversal of soil moisture over short distances. In the following section, important criteria that should be considered by the designers of a monitoring WSN in small plots is described, which could be useful to determine the number of sensors to be used in the perfect grid methodology.

## 4. Analysis of the Tests and Selection Criteria for The Number of Nodes

In accordance with the results of the experiments conducted, the average error for each number of n×n nodes was determined. As mentioned before, the error decayed as the number of nodes increased. Therefore, people who decide to use a WSN in small plot can take into account the results of our experiments to find a proportionality between the error to assume in the implementation and the number of nodes to be used. It is important to remember the that the cost of construction of each node is approximately 160 dollars; each node is composed of a battery, a solar panel, a wooden frame for support, a box for protection against water, a microcontroller, a humidity sensor, an Xbee module for communication, terminals, connection lines, a regulator for the panel and connectors.

Using a reference map with the 289 nodes strategy would require an overall investment of $46,240 dollars. As the objective is to reduce the number of nodes and thus the implementation costs, being able to the error in the spatial representation of the humidity within the crop can help reduce costs. Figure 19 shows the error found in each of the n×n networks and its relation to the implementation costs of the network.

Figure 19 shows the error in the third test, only as a reference for the way in which the number of nodes could be selected in a crop of 8 m × 8 m. When compared with the cost of implementation, a point at which the error tends to stabilize was found compared to the reference map. It is at this point that it is necessary to decide whether to increase the investment to reduce this error. With the node design used in our research, for n=5, the error oscillates by around 1.4% of soil moisture in the entire crop, which may not be very significant in terms of the initial reference map. It is important to note that if the implementation costs are reduced, the positive slope presented in the cost could be reduced. However, a methodology has been presented to select the number of nodes needed for a small plot in a potato crop. If the farmer or researchers decided to implement this methodology, they would have a reference to spatially analyze soil moisture behavior. For any of the n×n configurations of n=2,…,11, we have shown that the number of initial 289 nodes can be reduced to a low number for implementation.

## 5. Conclusions and Future Work

This work presented a method for the estimation of soil moisture by means of a perfect grid approach in a potato crop, achieving low errors in comparison to a reference map. The proposed method allows a good characterization of the behavior of soil moisture. In addition, it also functioned as a strategy for selecting the number of nodes to be used in a WSN, in such a way that the behavior of moisture can be characterized.

The proposed methodology focused on optimal nodes distribution diverges from most of the existing methods in the state of the art for temporary estimation of soil moisture, in which the locations of the nodes are at strategic points, in accordance with the topography (commonly used at large areas), or those positioning the nodes randomly but having reservations about the optimal number of nodes to be used (at small crops).

This study also presented a methodology that is helpful when evaluating the number of nodes to be used and the cost of implementation, providing an answer to the optimal number of sensors that would be necessary to accurately represent the behavior of soil moisture in a crop. Combining the perfect grid and the evaluating chart, it gives an optimal implementation design, in terms of cost and margin of error. However, it should be noted that the final decision to determine the optimal number of sensors within the crop will depend on the farmer or researcher, in accordance with their main goal or resources.

Regarding the implementation presented in this work, if we compare the use of 289 nodes initially with an estimated implementation cost of around $46,000 dollars (and very low estimate errors), we have met the goals of reducing the number of nodes, keeping the error to an acceptable low value and developing a map of 25 nodes for a potato crop. The implementation costs were reduced to $4000 dollars, which means a decrease of the initial cost by 11.5 times. Future work will be focused on reducing the implementation costs per node, and improving the estimation algorithms at the edges of the crop, in hopes of further reducing the error.

For future works it would be relevant to explore the use of convolutional networks in the WSN to optimize the interpolation algorithm. Additionally, it would be useful to analyze the possible benefits of storing the maps in the cloud through IoT. It allows one to contrast one’s data with other crop maps (i.e., the Crop Water Stress Index (CWSI) and other soil moisture maps generated in the same area or region) temporally and spatially, which would be useful for correlating with available seasonal water resources used for crop irrigation. Additionally, with a correlation between the soil moisture maps of the perfect grid and data obtained using multispectral imaging orthomosaics when deploying unmanned aerial vehicles (UAVs), a fast and affordable monitoring system, could be also possible.

## Figures and Tables

**Figure 1 plants-11-01636-f001:**
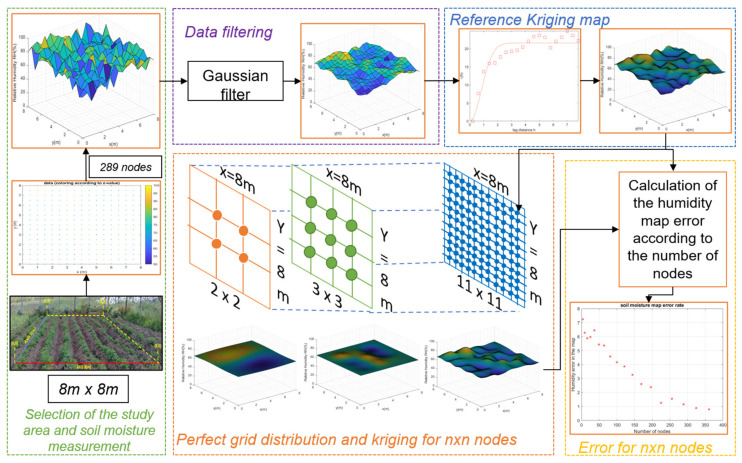
Proposed architecture for soil moisture estimation and optimal node positioning.

**Figure 2 plants-11-01636-f002:**
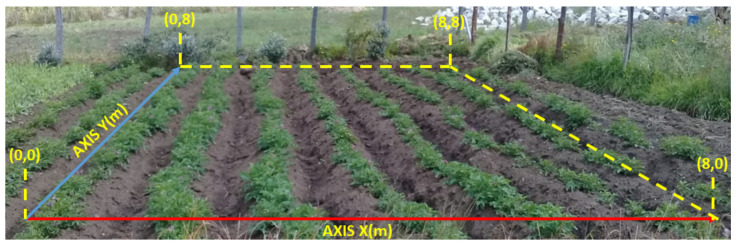
Small experimental plot with potatoes (8 m × 8 m), Solanum Phureja variety, 60 days after germination.

**Figure 3 plants-11-01636-f003:**
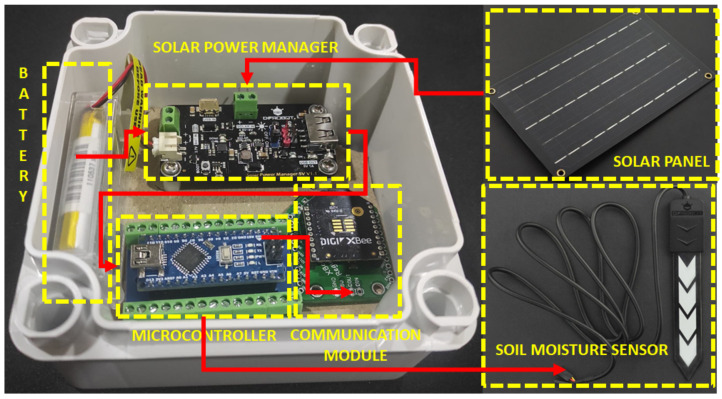
The internal components that make up each of the nodes of the WSN network: power supply (battery, solar panel and its controller), microcontroller, communication system, humidity sensor.

**Figure 4 plants-11-01636-f004:**
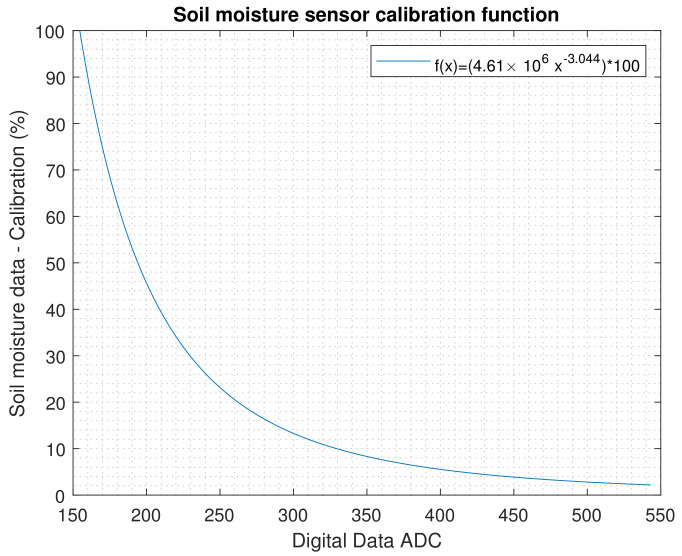
Calibration curve of the moisture sensor resulting from the gravimetric method; this calibration function is located inside the microcontroller of each node.

**Figure 5 plants-11-01636-f005:**
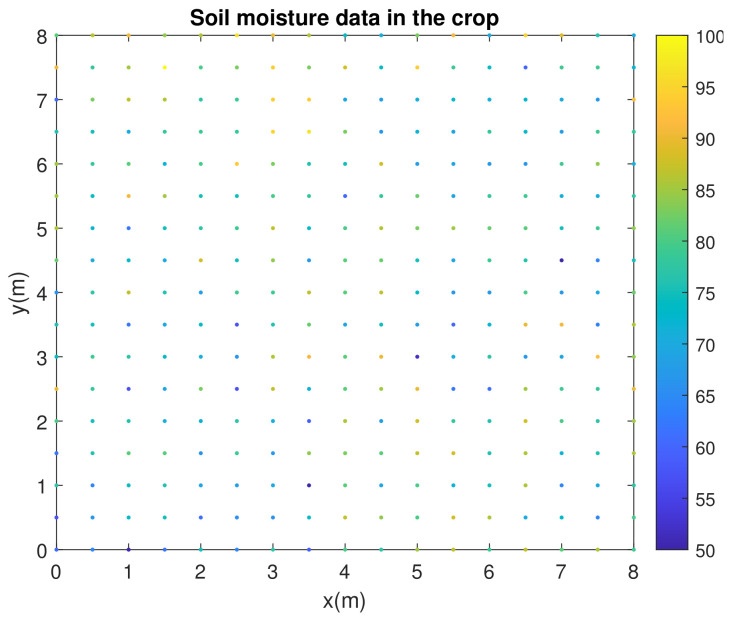
Two-hundred and eighty-nine measurement points within the experimental crop (8 m × 8 m) to generate the reference soil moisture map.

**Figure 6 plants-11-01636-f006:**
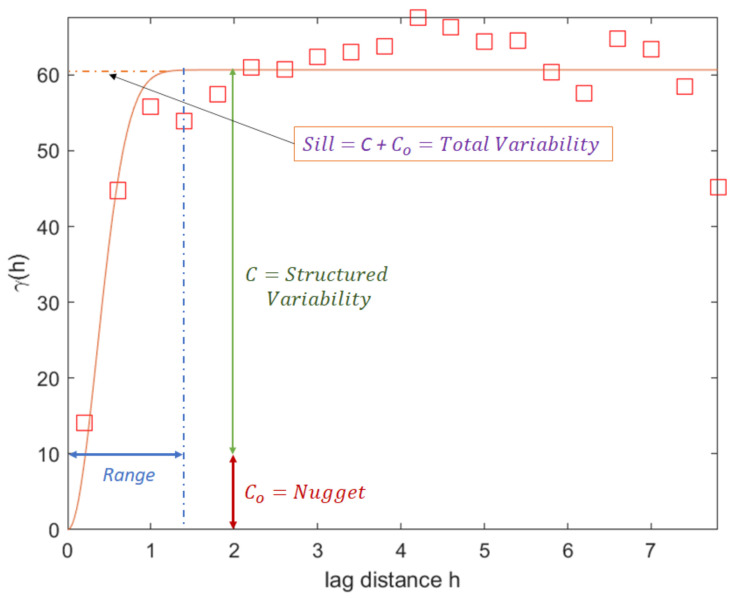
Graphical description of the variogram (variation in relation to distance between two points) and its components: range, nugget and sill.

**Figure 7 plants-11-01636-f007:**
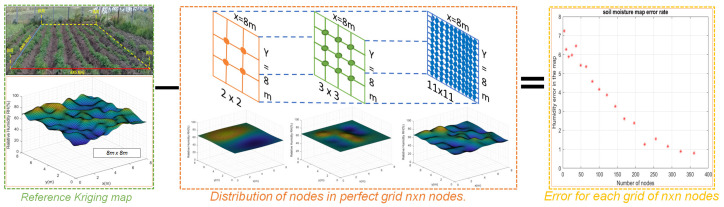
Proposed methodology for the calculation of the error of the n×n nodes with reference to the initial reference map.

**Figure 8 plants-11-01636-f008:**
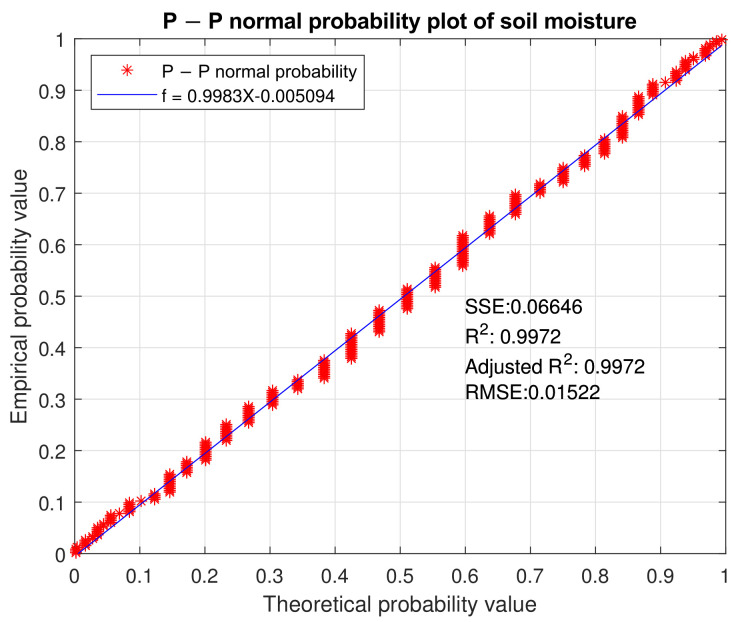
P-P normal probability plot of soil moisture.

**Figure 9 plants-11-01636-f009:**
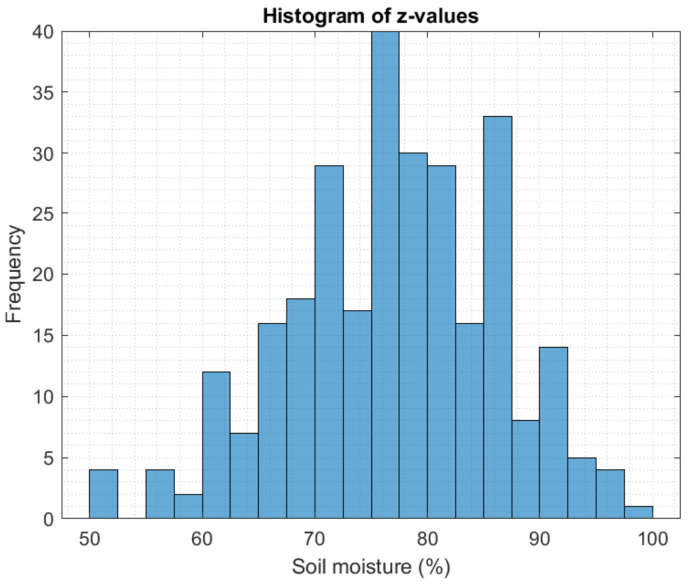
Example histogram of the initial soil moisture of the potato crop.

**Figure 10 plants-11-01636-f010:**
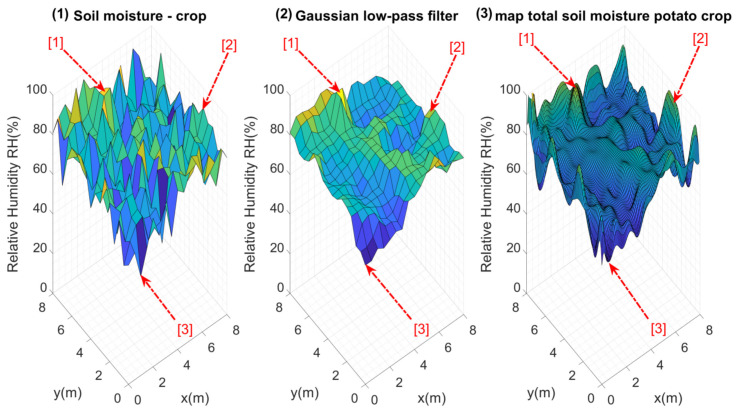
(**1**) Original data obtained from the 289 points measured within the crop, (**2**) data after the data smoothing filter and (**3**) Kriging map that served as the initial reference map. Points [1], [2] and [3] within the figure show the maximum and minimum points detected in some regions.

**Figure 11 plants-11-01636-f011:**
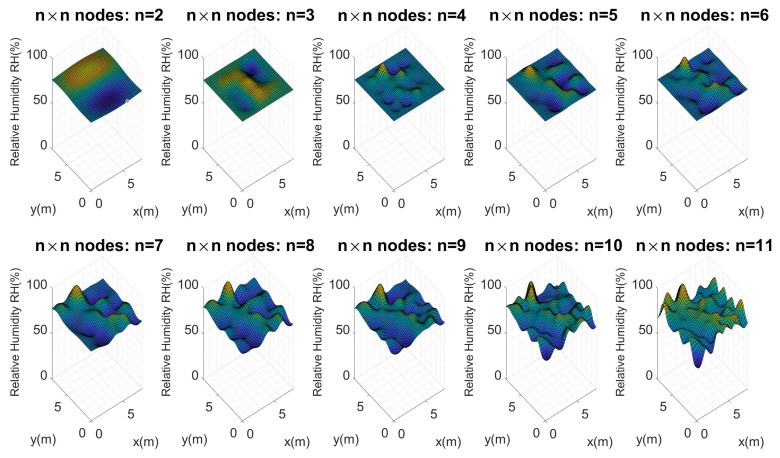
Soil moisture distribution maps in the test crop using n×n nodes. First experimental test.

**Figure 12 plants-11-01636-f012:**
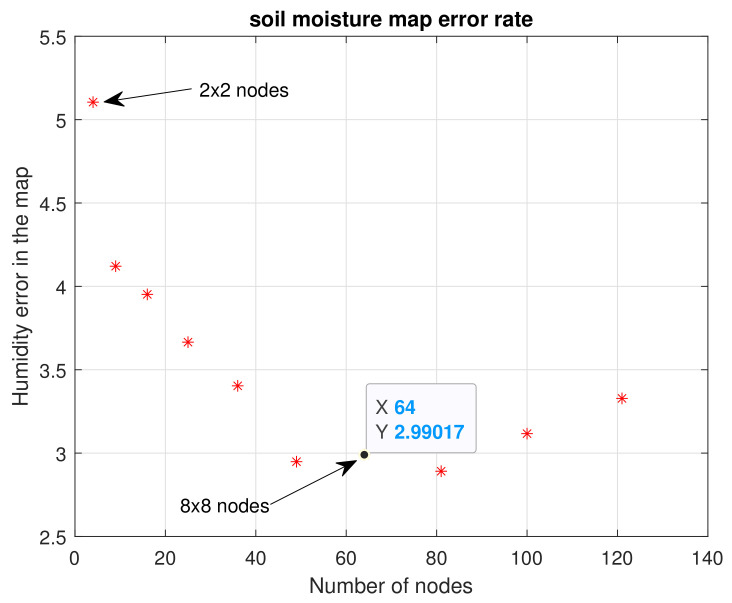
Error encountered for n×n nodes. First experimental test.

**Figure 13 plants-11-01636-f013:**
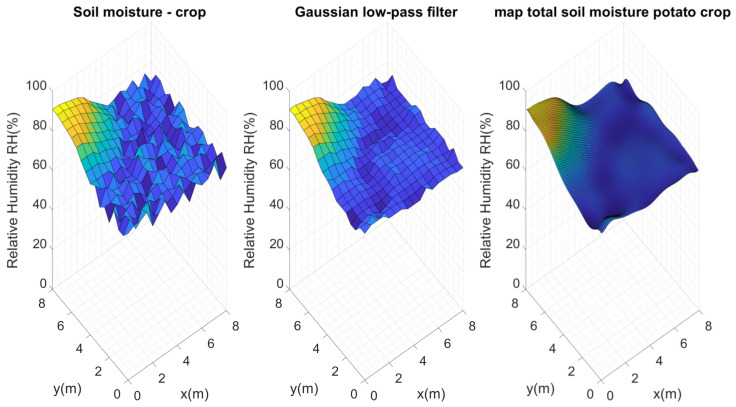
Irrigation focused on one end of the crop.

**Figure 14 plants-11-01636-f014:**
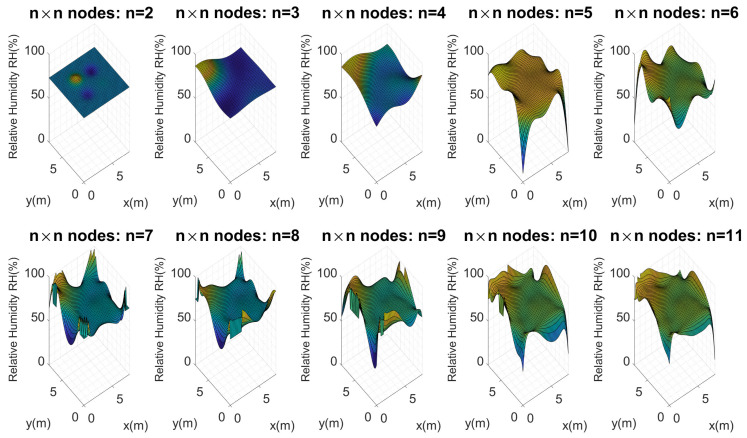
Soil moisture distribution maps in the test crop using n×n nodes. Second experimental test.

**Figure 15 plants-11-01636-f015:**
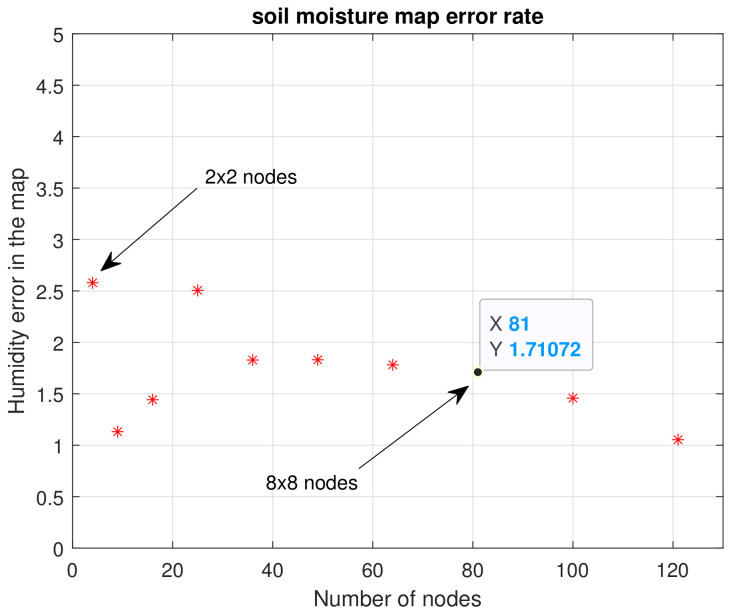
Error encountered for n×n nodes. Second experimental test.

**Figure 16 plants-11-01636-f016:**
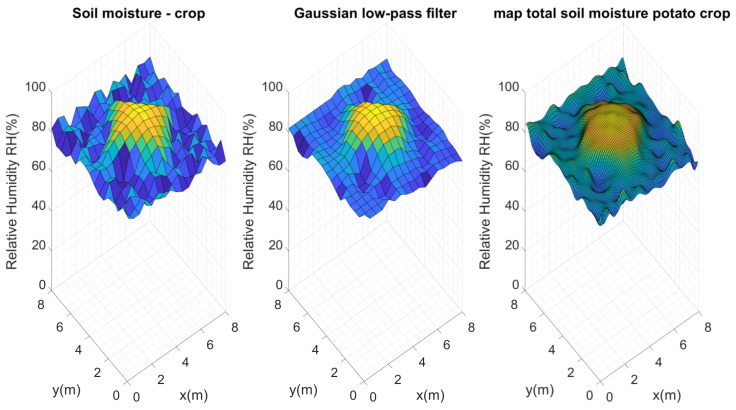
Soil moisture reference map after irrigation in the center of the crop.

**Figure 17 plants-11-01636-f017:**
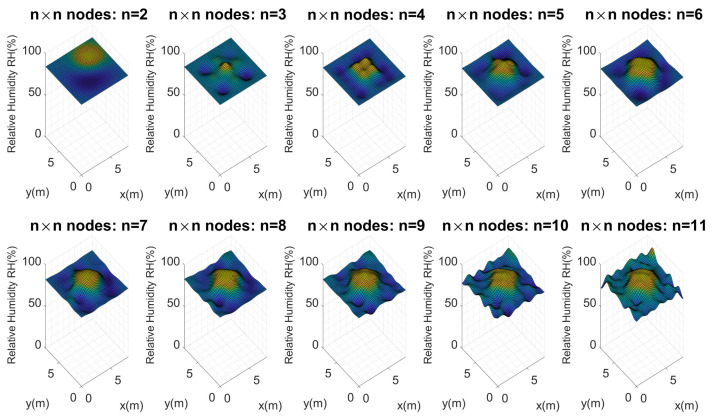
Soil moisture distribution maps in the test crop using n×n nodes. Third experimental test.

**Figure 18 plants-11-01636-f018:**
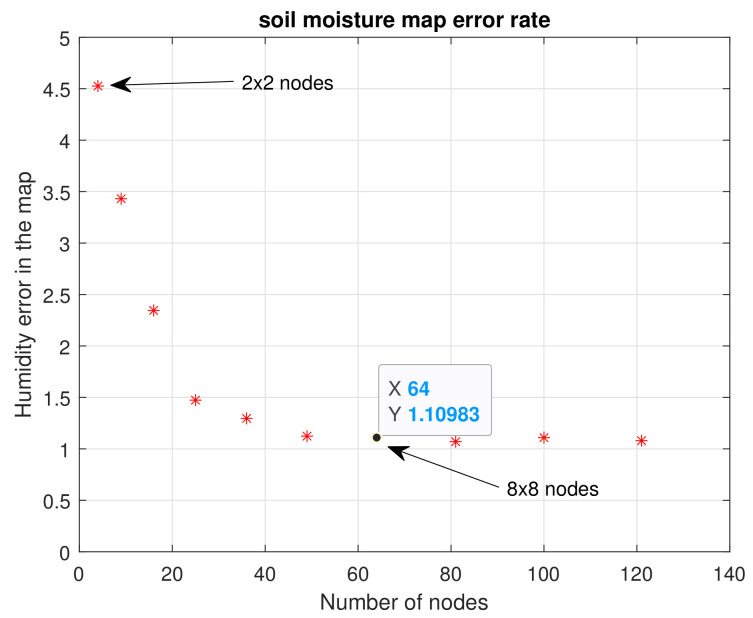
Error encountered for n×n nodes. Third experimental test.

**Figure 19 plants-11-01636-f019:**
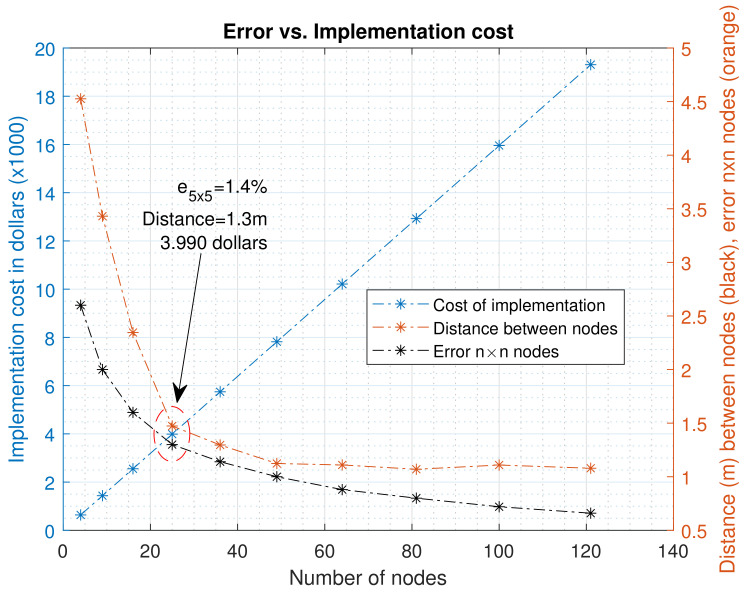
Relationship between the number of nodes, the cost of implementation, the error of the maps for n×n nodes and the distance between nodes for a potato crop of 8 m × 8 m.

**Table 1 plants-11-01636-t001:** Skewness and kurtosis of the samples.

Number of Samples	$\bar{z}$	σ	$s^{2}$	C.V	$Z_{\max}$	$Z_{\min}$
289	76.75	9.2515	85.59	12.05	100	50

**Table 2 plants-11-01636-t002:** Distance between nodes for n×n, for n=2,…,11.

**Number of Nodes**	**4 Nodes**	**9 Nodes**	**16 Nodes**	**25 Nodes**	**36 Nodes**
Distance (m)	2.6	2	1.6	1.3	1.14
**Number of Nodes**	**49 Nodes**	**64 Nodes**	**81 Nodes**	**100 Nodes**	**121 Nodes**
Distance (m)	1	0.88	0.8	0.72	0.66

## Data Availability

The data used to support the findings of this study are available from the corresponding author upon request.

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
