# Peer review of "Optimal Deployment of WSN Nodes for Crop Monitoring Based on Geostatistical Interpolations"

_plants, 2022, doi:10.3390/plants11131636_

Round 1

Reviewer 1 Report

In this paper, the authors proposed an optimal approach forWSN deployment by using the kriging interpolation technique. After the experiments with 289 sensors, they concluded that 25 sensors are good enough, which is very interested for me. Overall, this paper is well written, here are two minor suggestions:

1. All the equations are part of the paragraph, therefore, I would recommend adding a comma at the end of each equation.

2. Line 204, for "R2", "2" should be superscript.

Author Response

Thank you for your comments. Attached is the PDF file with the response to your comments and suggestions.

Reviewer 2 Report

In the manuscript titled “Optimal deployment of WSN nodes for crop monitoring based on geostatistical interpolations”, the authors reported an interesting and valuable study. And this study could provide some implications for others researchers. However, there are some major drawbacks in the study which need to be addressed before the manuscript could be consider publication on Plants. Based on these I would like to suggest reconsider after major revision. The detailed comments are as follow:

1. Abstract needs to be supplemented with research background and conclusions.

2. I suggest adding kriging interpolation to keywords.

3. The significance of predicting soil moisture should be described in the introduction.

4. The introduction lacks a description of the shortcomings of previous studies.

5. The expressions of previous research progress require revision. For example, in line 30, Fan et al developed a hydrological agricultural system based on kriging technology driven WSN operation.

6. I advise that Table 1 shows the skewness and kurtosis of the samples.

7. The font size of Figure11-16 could be larger.

8. I propose to provide a legend for the three line segments in Figure 19.

Author Response

(The authors gave the same response as above.)
